# HAPPI: Hyperbolic Hierarchical Part Prototypes for Image Recognition

Hooman Vaseli*    Victoria Wu*    Nima Kondori    Nguyen Nhat Minh To

Andrea Fung    Ang Nan Gu    Purang Abolmaesumi

The University of British Columbia, Vancouver, Canada

hoomanv@ece.ubc.ca, victoriawu@ece.ubc.ca, purang@ece.ubc.ca

## Abstract

*Prototypical part networks have gained prominence in computer vision due to their inherent interpretability, enabling decisions based on representative part features without post-hoc explanations. However, existing prototypical networks learn part-based features in flat Euclidean space, yet they could better capture the natural hierarchical relationships within image features to enhance performance on tasks requiring structural understanding. To address this opportunity, we propose HAPPI (**H**ierarchical **A**nd **P**art **P**rototypical **I**mage recognition), a framework that leverages hyperbolic geometry to organize prototypical part features hierarchically within a Lorentzian manifold. By arranging localized generic features near the hyperboloid origin and broader specific features farther away, HAPPI learns generic prototypes for defining local patterns and specific prototypes that aggregate broader discriminative cues, effectively capturing hierarchy in image data. Our approach is model-agnostic and can be applied to various prototypical neural networks and backbones. We evaluate HAPPI on several baselines and datasets, showing that hyperbolic prototypes match or outperform Euclidean ones while adding interpretability. Qualitative results reveal that generic prototypes highlight localized, class-defining traits, while specific prototypes capture broader patterns across larger regions, enabling differentiation through both local and contextual features. Our code can be found at*
*http://github.com/DeepRCL/HAPPI.*

## 1. Introduction

Prototypical part networks have emerged as a prominent field in computer vision due to their ability to make interpretable and explainable decisions. They hold an advantage over traditional black-box models as they can provide the reasoning behind their results without reliance on post-hoc explainability methods, leading to more trustworthy deci-

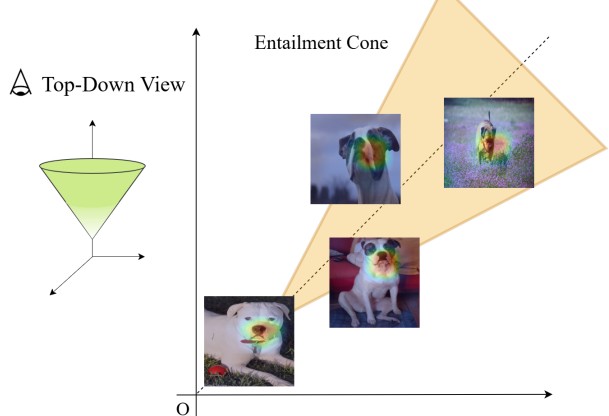

Figure 1. Organization of hierarchical part prototypes in hyperbolic space, where generic part prototypes, that capture localized, class-defining traits, entail specific part prototypes that capture broader contextual variations across larger regions.

sion making. Models such as ProtoPNet [2], ProtoPNet variants [4, 27] and PIP-Net [24] have shown success in a variety of image classification tasks.

While effective, existing prototypical part networks learn each prototype in a flat embedding space, without explicitly modeling the inherent hierarchical relationships between part features at different scales This can limit performance on tasks needing deeper structural understanding, such as object recognition, scene understanding, or fine-grained classification. In real-world images, features are naturally organized hierarchically, where generic features capture localized class-defining characteristics, while specific features aggregate broader patterns that help distinguish between similar classes. Generic features, through their localized focus on key class characteristics, provide clear discriminative signals for initial class separation. However, when differentiating between visually similar classes—such as distinguishing between dog breeds—specific features become more valuable, as they aggregate broader contextual

---

*Equal contribution

patterns necessary for precise classification. Therefore, by neglecting this hierarchy, existing prototypical models may miss critical context needed for accurate predictions.

As illustrated in Fig. 1, organizing prototypes hierarchically allows models to capture both localized and broader class features in a structured manner. Generic prototypes focus on fundamental and consistent traits that reliably characterize a class—for instance, in animal images, they may capture the typical shape of a nose, a paw, or an ear that remains stable across variations. In contrast, specific prototypes aggregate broader patterns by attending to larger regions of the image, combining multiple features and their relationships across wider areas (e.g., the overall body structure, or patterns spanning multiple body parts). By structuring part prototypes this way, the model can differentiate classes based on both their localized distinctive traits and the broader contextual patterns that help distinguish similar-looking instances.

To address this opportunity, we introduce **HAPPI**, a **H**ierarchical **A**nd **P**art **P**rototypical **I**mage recognition methodology, which learns part prototypes in a hyperbolic space using a Lorentzian manifold instead of Euclidean space. Our method structures the embedding space so that localized generic features—representing key class-defining characteristics—are positioned near the hyperboloid origin, while broader specific features—aggregating patterns across larger regions—are positioned farther away. This arrangement enables the model to learn generic part prototypes that focus on distinctive local traits, while specific part prototypes capture broader contextual patterns that help distinguish between similar classes. By explicitly modeling this hierarchy, our approach enhances the model's ability to differentiate between classes through both localized and broader-scale features. While demonstrated on several backbones, our approach has the potential to generalize to other prototypical neural networks and image classifiers. Our contributions are as follows:

- We propose a model-agnostic framework to optimize part-based prototypical networks on the Lorentzian manifold in hyperbolic space. We explicitly learn localized generic and broader specific prototypes while ensuring hierarchical consistency among prototypes of each class.
- Extensive experiments across various prototypical architectures and datasets show our hyperbolic approach matches or outperforms Euclidean counterparts while structuring prototypes into a more interpretable hierarchy.

## 2. Related Works

### 2.1. Prototypical Part Networks

Prototypical part networks are inherently interpretable models due to their structure. These models evaluate inputs explicitly based on their similarity to learned discriminative features, or "part prototypes," for each class. By visualizing the appearance and location of prototypes, as well as the coefficients relating input-prototype similarity to class logits, users can validate the model's decision-making process. ProtoPNet [2] introduced classification using representative part prototypes for each class in the training set and providing visual validation by highlighting feature proximity and localization within input images.

Subsequent work has focused on enhancing interpretability and prototype efficiency. ProtoPool [28] and ProtoPShare [27] reduce the total number of prototypes by sharing them across classes. PIP-Net [24] adopts a self-supervised approach to produce sparse prototypes and human-understandable features, while XProtoNet [16] extends interpretability by allowing different prototype sizes, applying prototypical neural networks successfully to radiology datasets. SPANet [34] further improves explainability by combining part prototypes with semantic concepts, providing clearer interpretations of what each prototype represents. These approaches enhance interpretability by either structuring prototypes more efficiently or aligning them with semantically meaningful concepts, supporting more intuitive and transparent decision-making.

Other approaches have modified the learning formulation to improve performance. TesNet [37] employs a Grassmann manifold to create distinct class subspaces, and ST-ProtoPNet [36] introduces an SVM-like method to learn boundary-supporting prototypes. Prototypical networks have also been integrated into transformer architectures [5], as seen in ProtoPFormer [39], ProtoFormer [6], and most recently, ProtoViT [18]. ProtoTree [23] uses a decision-tree structure to reduce the number of prototype comparisons, enhancing classification efficiency. Recently, ProtoP-NeXt [38] demonstrated that cosine similarity and Bayesian tuning could improve ProtoPNet's transparency and performance across various architectures and classification methods.

Beyond interpretability and architectural improvements, recent works have explored hierarchical representations within prototypical part networks. MCPNet [35] and HPDR [12] introduce hierarchical prototypes, albeit in different ways. MCPNet learns hierarchical representations through multi-scale prototypes, while HPDR refines feature distributions using hierarchical part prototypes in hyperbolic space. These methods leverage hierarchy to better capture structural relationships within data, enhancing representation learning and classification performance.

### 2.2. Hierarchical Representations

Image datasets inherently exhibit diverse types of hierarchical relationships. One such hierarchy involves clear-ambiguous relationships, where clear images are associated with specific classes, while ambiguous images (e.g., blurred

or occluded) are more generic and tend to exhibit features that overlap with multiple classes. Another type of hierarchy is the whole-part relationship, where a global view captures whole objects or scenes, while local views focus on finer details of objects or scene parts. Each global view is associated with multiple local views, allowing local features to be contextualized within the broader scope of the image (e.g. a leaf is part of a tree and a tree is part of a park). Recognizing these hierarchical relationships is essential for understanding the intricate relationships within complex visual objects or scenes [15, 25].

Prior works have proposed explicit methods to represent hierarchical concepts within images. For example, PyramidCLIP [7] captured multi-granular image representations by learning at global (whole image), intermediate (large image patch), and local levels (cropped object images). This approach modeled relationships across these levels using self-supervised learning guided by language, resulting in more separable representation spaces and improved zero-shot classification.

## 2.3. Hyperbolic Learning

Hyperbolic learning has demonstrated strong potential for capturing hierarchical structures within data [21]. Unlike Euclidean space, which has a flat manifold, hyperbolic manifolds offer a curved geometry that naturally preserves hierarchical relationships. It can effectively represent tree-like structures where nodes at different levels capture features at different scales of abstraction. There are two main approaches to learning visual representations in hyperbolic space: the Poincaré model [13, 15, 20] and the Lorentzian model. The Poincaré model represents hyperbolic space as the interior of a disk (in 2D) or a ball (in higher dimensions) within Euclidean space. The Lorentzian model represents hyperbolic space through a hyperboloid manifold embedded in Minkowski space. Both models have been successfully applied in computer vision tasks. In our work, we adopt the Lorentzian model due to its numerical stability and optimization advantages in practice, following recent works like MERU [3] and HyCoClip [25].

Desai et al. [3] used the Lorentz model to enhance vision-language representation learning. Their proposed approach, MERU, learned a contrastive model (i.e. CLIP) in hyperbolic space where hierarchy in visual and language concepts was preserved. MERU adapted the contrastive loss to minimize the proximity between the associated image and text using Lorentzian distance instead of cosine similarity. Additionally, MERU proposed an entailment loss that enables the model to learn that text represents more abstract, generic concepts than corresponding images (e.g., text of 'dog' encompasses many dog images). The entailment loss pushes image embeddings related to a specific text to exist within a cone-shaped space emanating from the text em-

bedding, indicating that the text entails a set of associated images.

Inspired by MERU, our work, HAPPI, extends these ideas to prototypical part networks, tailoring the embedding space and loss functions for prototype-based learning. While prior work has explored hyperbolic prototypes [1, 8, 9, 15, 17], these approaches focus on class-level prototypes, whether learned or computed as class means. In contrast, HAPPI is the first method to learn feature-level (part-based) prototypes in hyperbolic space, where we learn multiple, spatially grounded prototypes per class that are structured hierarchically via Lorentzian entailment loss. This enables interpretable, part-based reasoning ('this part looks like that') not supported by single-prototype embedding methods, providing transparency in how model predictions are derived by showing which parts of training images contributed to the classification decision.

## 3. Method

### 3.1. Prototypical Classification Framework

Prototypical part networks classify inputs by measuring their similarity to class-specific part prototypes. Given an input $x$, a feature extractor $f(\cdot)$ generates a feature map $f(x) \in \mathbb{R}^{H \times W \times D}$, where $H$ and $W$ are spatial dimensions and $D$ is feature depth. These features are then processed for comparison with prototypes. Let $V_{\text{euc}} \in \mathbb{R}^D$ denote each extracted feature in Euclidean space to be compared with prototypes. Prototypes are denoted as $p_{c,k} \in \mathbb{R}^D$, where $c$ and $k$ represent the class and the index of the prototype within that class, respectively, with typically $K$ prototypes per class. The comparison between $V_{\text{euc}}$ and $p_{c,k}$ yields similarity scores $s(V_{\text{euc}}, p_{c,k})$, usually computed using Euclidean distance or cosine similarity. These scores are then fed into a fully connected layer $FC : \mathbb{R}^{K \times C} \to \mathbb{R}^C$, producing class prediction scores $\hat{y} \in \mathbb{R}^C$, where $C$ is the number of classes. This prototype-based approach enhances interpretability by basing classifications on similarities to known examples, facilitating clearer pattern identification in the data.

### 3.2. Transition to Hyperbolic Space

Our method extends traditional prototypical networks by projecting the prototypes and extracted features into hyperbolic space, specifically using the Lorentz model of the hyperboloid. Following MERU's successful approach to adapting CLIP to hyperbolic space [3], we maintain the backbone's feature extraction in Euclidean space and project these features to the Lorentzian manifold afterwards. This design choice allows us to leverage well-established CNN and Transformer architectures while gaining the benefits of hyperbolic geometry for prototype organization. In this model, the $V_{\text{euc}}$ is mapped to a $(D+1)$-

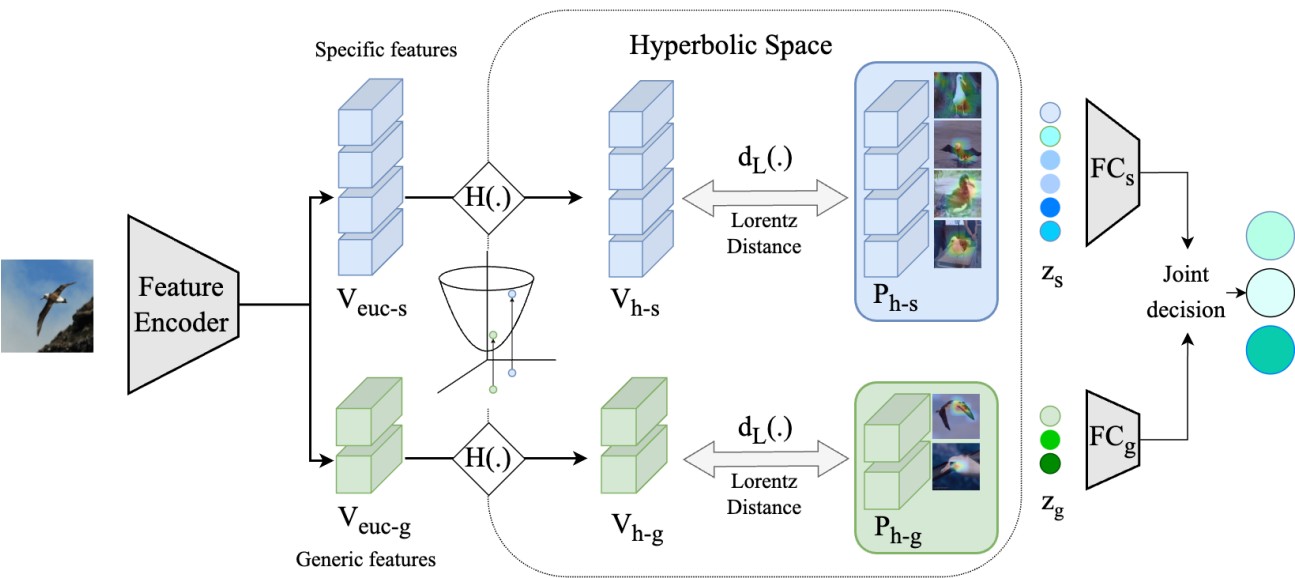

Figure 2. An overview of our proposed approach. We use a given prototypical network's feature encoder to extract both generic and specific features, then lift those features to the hyperboloid using exponential mapping. The lifted features are then compared to their respective prototypes, and similarity scores are generated based on their distances to the prototypes. These similarity scores ultimately result in the activations that are averaged to create the class logits.

dimensional hyperbolic space by adding an additional time dimension. This mapping results in a hyperbolic feature $V_h = [V_{\text{time}}, V_{\text{space}}]$, where $V_{\text{space}} \in \mathbb{R}^D$ represents the spatial components and $V_{\text{time}} \in \mathbb{R}$ is the time-like dimension [22]. This representation can capture hierarchical relationships and varying levels of detail more effectively than Euclidean space.

To obtain the $V_{\text{space}}$, a mapping operation, shown as $\mathcal{H}(.)$ in Figure 2, projects Euclidean space features $V_{\text{euc}}$ onto the hyperboloid. In our case, we use the simplifying assumption from [3] which considers only the exponential map centered at the origin. Under these conditions, the map simplifies to:

$$V_{\text{space}} = \frac{\sinh\left(\sqrt{c}\|V_{\text{euc}}\|\right)}{\sqrt{c}\|V_{\text{euc}}\|} V_{\text{euc}} \qquad (1)$$

where $c$ is the negative curvature of the hyperbolic space. Since the hyperbolic features are constrained to reside on the hyperboloid manifold, the time dimension $V_{\text{time}}$ can be derived based on $V_{\text{space}}$ as follows:

$$V_{\text{time}} = \sqrt{\frac{1}{c} + \|V_{\text{space}}\|^2} \qquad (2)$$

As a result, we can effectively map the $V_{\text{euc}}$ from Euclidean space into the hyperboloid by calculating both $V_{\text{space}}$ and $V_{\text{time}}$. We can then calculate the Lorentzian distance of the resulting features from the learnable prototypes to identify the most prominent prototypes and classify the input image. For example, let $a = [a_{\text{time}}, a_{\text{space}}]$ and $b = [b_{\text{time}}, b_{\text{space}}]$

denote two points on the hyperboloid. The Lorentzian distance $d_L(a, b)$ is defined as:

$$d_L(a, b) = \frac{1}{\sqrt{c}} \cosh^{-1}(-c\langle a, b\rangle_L), \qquad (3)$$

where $\langle \cdot, \cdot \rangle_L$ denotes the Lorentzian inner product. This inner product is defined as:

$$\langle a, b \rangle_L = \langle a_{\text{space}}, b_{\text{space}} \rangle - a_{\text{time}} b_{\text{time}}, \qquad (4)$$

where $\langle \cdot, \cdot \rangle$ is the standard Euclidean inner product. This distance metric is foundational in aligning the prototypes with features in the embedding space.

### 3.3. Hyperbolic-Based Losses and Prototypes

In this section, we introduce the adaptation of prototypical losses and prototype structures to hyperbolic space. We describe the conversion of Euclidean-based clustering and separation losses to their hyperbolic counterparts and introduce generic prototypes for capturing defining and consistent features. We also present an entailment loss inspired by MERU [3] to encourage hierarchical structure in hyperbolic space. Prototype optimization methods (such as clustering and separation losses [2]) are based on hyperbolic distances.

### 3.3.1. Feature Extraction and Prototype Assignment

We introduce two types of prototypes per class: generic prototypes, which capture localized, distinctive features, and specific prototypes, which aggregate broader contextual patterns across larger regions of the image. Generic

features are extracted explicitly according to the backbone architecture:

- Transformer-based: the [CLS] token is used to represent generic features, following ProtoPFormer [39].
- CNN-based: Attention maps, interpreted as occurrence maps, are applied to extract generic features, inspired by XProtoNet [16].

Specific features, in contrast, are taken directly from the host architecture's feature extraction pipeline without additional processing. without additional processing.

For both feature types, we dedicate corresponding prototypes: generic and specific prototypes. Once extracted, we utilize Equations 1 and 2 to project both generic ($V_{\text{euc-g}}$) and specific ($V_{\text{euc-s}}$) features to the hyperbolic space, forming $V_{\text{h-g}}$ and $V_{\text{h-s}}$, respectively. Their corresponding prototypes are also projected, resulting in hyperbolic generic prototypes $P_{\text{h-g}}$ and hyperbolic specific prototypes $P_{\text{h-s}}$.

### 3.3.2. Prototype-based Classification

Prototype-based classification leverages both generic and specific prototypes to enhance predictive accuracy. We compute the similarities of extracted features to both sets of prototypes. To obtain the final class activation logits, we calculate two separate sets of logits based on these similarities. Each set of similarities is passed through a distinct final layer to produce corresponding logits: $z_s$ for specific prototypes and $z_g$ for generic prototypes. The final activation is a joint decision that is computed as the average of these two sets of logits. This approach allows the model to integrate both generic and specific information for classification, improving its ability to perform effectively across varying levels of granularity.

### 3.3.3. Clustering and Separation Losses

Clustering and separation losses are designed to encourage the learning of effective prototypes that capture the structure of the data. Clustering loss promotes intra-class compactness by encouraging features to be close to the prototypes of their corresponding classes, while separation loss enhances inter-class separability by pushing features away from prototypes of different classes [2]. Similar to existing prototypical approaches, we cluster extracted specific and generic features that are mapped to the hyperboloid, $V_{\text{h}}$, with the prototypes of the corresponding classes. Likewise, we separate the lifted features from the prototypes of different classes, using the following equations:

$$\mathcal{L}_{\text{clst}} = \min_i d_L(V_t^i, P_i^y), \quad \mathcal{L}_{\text{sep}} = -\mathbb{E}_{c \neq y, i}\left[d_L(V_t), P_i^c)\right],$$
(5)

where $V_t$ represents generic ($g$) or specific ($s$) feature embeddings, and $P_i^c$ denotes class-specific prototypes. Both clustering and separation criteria, denoted for specific prototypes as $L_{\text{clst}}^{\text{specific}}$ and $L_{\text{sep}}^{\text{specific}}$, and generically as $L_{\text{cluster}}^{\text{generic}}$ and $L_{\text{sep}}^{\text{generic}}$ respectively, utilize the Lorentzian distance,

Equation 3, as $d_L(V_{\text{h}}, p)$ where $p$ represents the prototype of interest. This adaptation maintains clustering and separation fidelity in hyperbolic space.

### 3.3.4. Entailment Loss

Entailment loss is designed to encourage a hierarchical structure in the hyperbolic space by enforcing relationships between specific and generic prototypes. It ensures that each specific prototype is entailed by at least one generic prototype of its class, creating a coherent hierarchy of representations. To encourage this hierarchical encoding, we implement an entailment loss based on MERU's formulation [3]. This is achieved by minimizing the exterior angle of $\text{ext}(V_{\text{h-g}}, V_{\text{h-s}})$ relative to the half-aperture $\text{aper}(V_{\text{h-g}})$:

$$L_{\text{entail}}(V_{\text{h-g}}, V_{\text{h-s}}) = \max(0, \text{ext}(V_{\text{h-g}}, V_{\text{h-s}}) - \text{aper}(V_{\text{h-g}})).$$
(6)

The half-aperture of a cone for entailment relationships is calculated as:

$$\text{aper}(a) = \sin^{-1}\left(\frac{2R}{\sqrt{c}\|a_{\text{space}}\|}\right),$$
(7)

where $R$ is a constant that controls the boundary conditions near the origin. Furthermore, the exterior angle between two points on the hyperboloid is given by:

$$\text{ext}(a, b) = \cos^{-1}\left(\frac{b_{\text{time}} + a_{\text{time}} \cdot c\langle a, b\rangle_L}{\|a_{\text{space}}\|\sqrt{(c\langle a, b\rangle_L)^2 - 1}}\right).$$
(8)

### 3.3.5. Combined Loss Function

The overall loss function $\mathcal{L}$ uses cross entropy, $\mathcal{L}_{CE}$, for classification and integrates generic and specific clustering and separation losses, entailment loss as shown below:

$$\mathcal{L} = \mathcal{L}_{\text{CE}} + \lambda_{\text{clst-g}}\mathcal{L}_{\text{clst}}^{\text{generic}} + \lambda_{\text{sep-g}}\mathcal{L}_{\text{sep}}^{\text{generic}}$$
$$+ \lambda_{\text{clst-s}}\mathcal{L}_{\text{clst}}^{\text{specific}} + \lambda_{\text{sep-s}}\mathcal{L}_{\text{sep}}^{\text{specific}} + \lambda_{\text{entail}}\mathcal{L}_{\text{entail}}$$
(9)

where each $\lambda$ denotes the coefficient corresponding to each loss value.

## 4. Experiments

### 4.1. Datasets

We used two different datasets for our evaluations: CUB-200-2011 and Oxford-IIIT Pets.

- **CUB-200-2011 (Caltech-UCSD Birds 200)** [33]: The CUB-200-2011 dataset is a widely-used benchmark for fine-grained classification tasks, focusing specifically on bird species recognition. It consists of 11,788 images across 200 bird species, each with a varying number of images. The dataset is split into 5,994 images for training and 5,794 images for testing.

- **Oxford-IIIT Pets** [26]: The Oxford-IIIT Pets dataset is used for pet breed classification and includes a balanced mix of cat and dog breeds. It contains 7,349 images spanning 37 breeds, with around 200 images per breed. The dataset is split into 3,680 images for training and 3,669 images for testing.

While each of these datasets has additional attribute annotations such as bounding boxes or segmentations, we only used the labels to train our models. We selected these datasets for their frequent use in prototypical neural network research and their diversity: CUB includes 200 classes, while Pets includes 37 classes.

## 4.2. Implementation Details

To ensure a comprehensive evaluation, we selected a range of backbone architectures that cover diverse structural approaches. Specifically, we included part-based prototypical neural networks like ProtoPNet and ST-ProtoPNet, which use patch-based prototypes; XProtoNet, which uses adaptable prototype sizes; and ProtoPFormer, which uses transformer-based backbone. For CNN-based evaluations, all models were tested with a ResNet-50 backbone [10], while different transformer-based models were also evaluated to assess performance across architectural types. Additionally, a black-box version of the backbone classification model was evaluated for comparison, as shown in the results section. We also evaluated MCPNet [35], a state-of-the-art hierarchical prototypical model in Euclidean space, and PIPNet [24], a state-of-the-art prototypical model. All evaluations were performed using 10 specific prototypes per class, a standard choice in prototypical neural network literature, along with 2 generic prototypes per class, determined empirically. The entailment loss coefficient, $\lambda_{entail}$, was set to 0.1 and R was set to 0.1. The loss weights, $\lambda$, for the remaining components were selected to match the values used in the baseline model. Input images were resized to 224x224, with shear and flip transformations for data augmentation, while test images were resized to 224x224 without additional cropping to maintain consistency.

For ProtoPFormer, we adopted their approach of using the CLS token as a generic prototype and image tokens as specific prototypes. Instead of ProtoPFormer's prototypical part concentration (PPC) loss, we implemented our clustering and separation loss functions. While PPC loss focuses on concentrating prototypes on distinct, centralized representative parts for each class, our clustering and separation losses also enforce the learning of distinct and representative prototypes, with a formulation that translates more effectively to hyperbolic space. For ProtoPNet and XProtoNet, we report two accuracy metrics: end-to-end (E2E) training accuracy where the prototypes and features are optimized jointly, and multi-stage training accuracy. Multi-stage training consists of initial prototype warm-up epochs,

followed by joint training, and then final layer optimization for additional epochs. To ensure a fair comparison, we retained the original backbone feature learning rates from each model's source paper. Further implementation details can be found in the supplementary materials.

## 4.3. Quantitative Results

Our method delivers strong performance across prototypical architectures, often matching or surpassing baselines. For fair comparison, all methods are implemented with consistent data preprocessing adopted from PipNet [24], though this leads to some differences from originally reported results. As shown in Table 1, HAPPI substantially improves performance on simpler architectures: on CUB-200-2011, ProtoPNet improves from 45.24% to 61.44%, while on Oxford-IIIT Pets it increases from 54.65% to 88.25%, gains of 16.2% and 33.6%, respectively. In contrast, more advanced models show smaller improvements: XProtoNet rises from 73.82% to 75.46%, ST-ProtoPNet from 86.54% to 87.21%, and ProtoPFormer backbones by only 0.5-4.0% on CUB-200-2011.

These differences may stem from the decision-making behavior of different architectures. Jiang et al. [14] show that transformers are more compositional, integrating evidence across regions, while standard CNNs behave more disjunctively, relying on localized cues. HAPPI therefore yields the largest gains on simple CNN-based ProtoPNet, where hierarchical hyperbolic prototypes complement the disjunctive behavior. In contrast, transformers already benefit from compositional reasoning, so their gains are smaller. Some CNN variants such as XProtoNet and ST-ProtoPNet introduce mechanisms (variable prototype sizes or support/trivial prototypes) that resemble this compositional behavior, explaining their more modest improvements. Still, HAPPI consistently improves interpretability by imposing a hierarchical structure where both generic and specific prototypes contribute to decision-making.

Despite these overall advantages, HAPPI does not always improve performance. For example, ProtoPFormer with a DeiT-Ti backbone on cropped CUB-200-2011 remains unchanged at 79.65%, likely due to limited model capacity. These outcomes suggest that the benefits of hyperbolic embeddings can be model- and dataset-dependent.

Compared to black-box baselines, HAPPI achieves competitive accuracy while significantly improving interpretability. On CUB-200-2011, the ResNet-50 black-box model attains 75.16% accuracy, whereas HAPPI-augmented ProtoPNet reaches 76.44%. On Oxford-IIIT Pets, hyperbolic XProtoNet achieves 91.50% accuracy, surpassing the ResNet-50 black-box model's 90.49%. These results highlight HAPPI's ability to balance high performance with transparency, offering a compelling alternative to opaque black-box models.

| Backbone | Methods | CUB | | CUB Cropped | | Pets | |
|---|---|---|---|---|---|---|---|
| | | Baseline (Euclidean) | HAPPI (Hyperbolic) | Baseline (Euclidean) | HAPPI (Hyperbolic) | Baseline (Euclidean) | HAPPI (Hyperbolic) |
| ResNet50 [10] | Black-box | 75.16 | - | 76.34 | - | 90.49 | - |
| | PIP-Net [24] | 73.52 | - | 82.00 | - | 88.50 | - |
| | MCPNet [35] | 74.28 | - | 70.78 | - | 90.32 | - |
| | ProtoPNet [2] | 45.24 | **61.44** | 62.30 | **80.51** | 54.65 | **88.25** |
| | ProtoPNet - E2E | 67.04 | **76.44** | 71.53 | **82.39** | 77.35 | **88.34** |
| | XProtoNet [16] | 73.82 | **75.46** | 80.84 | **81.03** | 89.48 | 89.40 |
| | XProtoNet - E2E | 75.87 | **77.91** | 82.09 | **82.90** | 90.00 | **91.50** |
| | ST-ProtoPNet [36] | 86.54 | **87.21** | 86.94 | 86.92 | 77.43 | **81.93** |
| DeiT-Ti [29] | Black-box | 75.27 | - | 78.13 | - | 88.85 | - |
| | ProtoPFormer [39] | 77.30 | 76.18 | 79.65 | 79.65 | 86.89 | 86.89 |
| DeiT-S [29] | Black-box | 79.86 | - | 82.53 | - | 92.23 | - |
| | ProtoPFormer [39] | 77.68 | **78.67** | 80.60 | **84.59** | 88.72 | **91.25** |
| CaiT-XXS-24 [30] | Black-box | 80.76 | - | 82.56 | - | 92.86 | - |
| | ProtoPFormer [39] | 80.86 | 80.81 | 82.24 | **83.91** | 91.47 | **91.88** |

Table 1. Classification accuracy (%) on full / cropped CUB-200-2011 and Oxford-IIIT Pets using various backbones and methods. All methods implemented with consistent data preprocessing for fair comparison.

## 4.4. Qualitative Results

To illustrate the qualitative effectiveness of HAPPI, we visualize individual learned prototypes in Figure 3, using the XProtoNet-E2E architecture on the Oxford-IIIT Pets [26] dataset. The figure includes four classes: Abyssinian (top left), Bengal (bottom left), American Bulldog (top right), and Basset Hound (bottom right). Each group has two rows—the bottom row shows a heatmap overlay highlighting attended regions, while the top row masks less relevant areas by shadowing them.

Generic prototypes capture consistent features that roughly distinguish classes. Across all classes, they focus on the nose and mouth area, a key trait distinguishing not only between cats and dogs but also among different breeds within each group. The American Bulldog's generic prototypes highlight facial wrinkles and white fur, while the Basset Hound's emphasize long ears and a brown head. The Abyssinian cat's prototypes focus on fur texture, nose, and eyes, while the Bengal cat's emphasize its distinct tiger-like fur pattern.

In contrast, specific prototypes capture broader context and variations in body shapes, poses, and colors. The American Bulldog's specific prototypes highlight a different color combination—white and brown—unlike its generic prototypes, which focus only on white fur. In the last specific prototypes of both the American Bulldog and Bengal cat, a diffused heatmap or less pronounced shadowing suggests a broader focus, capturing some background details.

Overall, generic prototypes identify localized, distinctive features, while specific prototypes aggregate broader patterns across larger regions, ensuring a hierarchical understanding from local to global scales. This complementary approach enables the model to differentiate classes through both fine-grained local characteristics and broader contextual patterns, leveraging information at multiple spatial scales.

## 4.5. Ablation Study

We performed an ablation study using XProtoNet-E2E to assess the impact of hyperbolic space and hierarchical structuring. As shown in Table 2, transitioning to hyperbolic space and introducing generic and specific prototypes improved performance over the Euclidean baseline, even without entailment loss ($\lambda_{entail} = 0$). Increasing $\lambda_{entail}$ slightly reduced accuracy as it constrains the embedding space to enforce hierarchical structure, where localized generic features must entail their corresponding broader specific features. This trade-off between accuracy and structural organization is consistent with observations in [3]. With $\lambda_{entail} = 0.1$, the model achieved a balance between maintaining high accuracy and ensuring meaningful hierarchical relationships between localized and broader features, making it our preferred setting.

To further analyze the effect of prototype allocation, we varied the number of generic ($K_g$) and specific ($K_s$) prototypes per class, as shown in Table 3. Increasing the number of prototypes generally improved performance, but gains plateaued around $K_g = 2$ and $K_s = 5$. Notably, models with more specific prototypes performed better than those with more generic ones, reinforcing the importance of capturing intra-class variability. The setup with $K_g = 5$ and

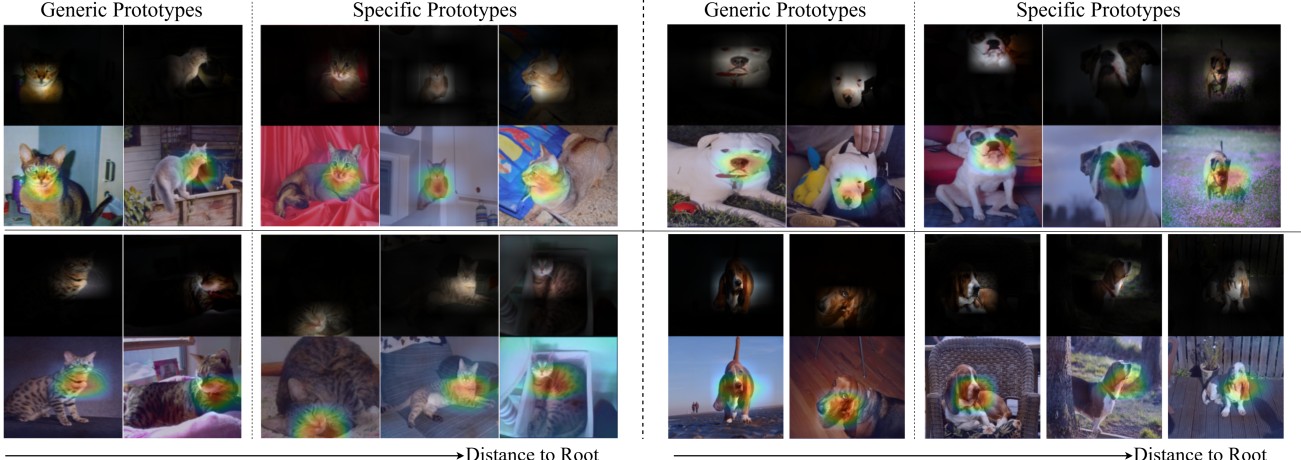

| Generic Prototypes | Specific Prototypes | Generic Prototypes | Specific Prototypes |

→Distance to Root    →Distance to Root

Figure 3. Qualitative visualization of learned prototypes in the XProtoNet-E2E architecture for the Oxford-IIIT Pets dataset [26]. Generic prototypes near the hyperboloid origin capture localized, distinctive features (e.g., specific facial features of a 'white American Bulldog'), while specific prototypes farther from the origin aggregate broader patterns across larger regions (e.g., overall body structure and patterns). As prototype distance from the hyperboloid root increases, features transition from localized characteristics to broader contextual patterns.

| Method | $\lambda_{\text{entail}}$ | | | |
|---|---|---|---|---|
| | 0 | 0.1 | 0.2 | 0.5 |
| Euclidean | 90.00 | - | - | - |
| Hyperbolic | 91.50 | 91.50 | 91.47 | 91.20 |

Table 2. Ablation study, using XProtoNet-E2E, on the use of hyperbolic space and entailment in terms of Accuracy (%) on the Pets dataset. The first row refers to the Euclidean model which has 10 prototypes per class, while the other rows are hyperbolic versions with 2 generic and 10 specific prototypes.

| $K_s$ | | 1 | 2 | 5 | 10 |
|---|---|---|---|---|---|
| $K_g$ | 1 | 88.83 | 90.27 | 91.06 | 91.06 |
| | 2 | 90.19 | 90.90 | 91.91 | 91.50 |
| | 5 | 90.79 | 88.53 | 91.77 | 91.52 |

Table 3. Accuracy (%) by number of generic and specific prototypes on the Pets dataset.

$K_s = 1$ performed worse than $K_g = 1$ and $K_s = 5$, indicating that generic prototypes alone are not sufficient for fine-grained classification. When specific prototypes were sufficient, reducing the number of generic prototypes from 5 to 2 further improved optimization stability. These results suggest that the ideal prototype configuration depends on the complexity of the dataset. A careful balance between generic and specific prototypes is crucial, as different datasets may require different levels of hierarchical structuring to achieve optimal performance.

## 5. Conclusion

In this paper, we introduced HAPPI, a model-agnostic approach for adapting prototypical part networks to hyperbolic space to effectively learn hierarchical representations from local to global scales. Our method showed comparable or improved performance over Euclidean baselines across diverse datasets while providing enhanced interpretability through its hierarchical structure. Qualitative analyses reveal that HAPPI organizes prototypes meaningfully, with localized generic prototypes capturing distinctive features near the hyperboloid origin and specific prototypes aggregating broader patterns farther away. Our findings suggest that hyperbolic prototypical networks, as exemplified by HAPPI, hold significant potential for applications requiring both multi-scale understanding and interpretability in visual tasks.

Future work could extend HAPPI in several directions. First, incorporating explicit label hierarchies (e.g., superclass–subclass structures in CUB or Pets) may help guide the embedding space, improving how generic and specific prototypes are learned, strengthening entailment, and further enhancing class separation. Second, our clustering and separation losses used the same coefficients as their Euclidean backbones; analyzing alternative weightings and running systematic sensitivity studies could yield better optimization and clarify robustness to hyperparameters. Third, applying HAPPI to larger hierarchical datasets such as iNaturalist [32] or TreeOfLife [11, 19] would test its scalability in settings with richer label structures. Finally, exploring adaptive prototype allocation and hyperbolic scaling strategies could tailor HAPPI more effectively to different backbones and datasets.

# 6. Acknowledgement

This work was supported in part by the Canadian Institutes of Health Research (CIHR), the Natural Sciences and Engineering Research Council of Canada (NSERC), and computational resources provided by Advanced Research Computing at the University of British Columbia [31].

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
