# HAPPI: Hyperbolic Hierarchical Part Prototypes for Image Recognition

## Supplementary Material

## A. Scaling Euclidean Features for Stable Hyperbolic Projection

In the MERU model [4], the extracted Euclidean features had an expected norm of $\sqrt{D}$ due to their CLIP-style layer initialization. This meant that when projected into hyperbolic space using the exponential map, their norm grew to approximately $e^{\sqrt{D}}$, which could cause numerical instability. To mitigate this, MERU applied a scaling strategy, introducing a learnable scalar $\alpha$, which was initialized as $\frac{1}{\sqrt{D}}$. This ensured that feature norms remained controlled after projection, preventing overflow issues in hyperbolic space.

However, this initialization does not generalize across architectures. The norm of extracted features is not inherently $\sqrt{D}$; instead, it depends on various factors such as the backbone network, layer configurations, and activation functions. In our case, the Euclidean feature norms do not follow the same distribution as in MERU, making the fixed $\frac{1}{\sqrt{D}}$ initialization unsuitable. Rather than assuming a predefined norm, we empirically estimate it by computing the mean norm of features extracted from the first batch of training data. Specifically, let $\mathbb{E}[\|V_{\text{euc}}\|]$ denote the average norm of Euclidean feature vectors in this initial batch. We then initialize the learnable scalar $\alpha$ as:

$$\alpha = \frac{1}{\mathbb{E}[\|V_{\text{euc}}\|]} \tag{10}$$

This ensures that feature norms remain controlled when mapped to hyperbolic space, mitigating numerical instability.

Furthermore, this same scaling approach cannot be directly applied to prototype vectors. Since prototype vectors are learnable parameters independent of the feature extraction process, their norms do not necessarily align with those of extracted features. To maintain consistency, we explicitly scale the prototype vectors in Euclidean space so that their mean norm matches the estimated mean norm $\mathbb{E}[\|V_{\text{euc}}\|]$. That is, before projecting prototypes into hyperbolic space, we rescale them such that:

$$\mathbb{E}[\|P_{\text{euc}}\|] = \mathbb{E}[\|V_{\text{euc}}\|] \tag{11}$$

where $P_{\text{euc}}$ represents the prototype vectors in Euclidean space.

By aligning the norm distributions of features and prototypes before projection, we ensure numerical stability while preserving a well-structured representation in hyperbolic space. This approach enables effective prototype-based classification without suffering from the norm explosion issues observed in prior work.

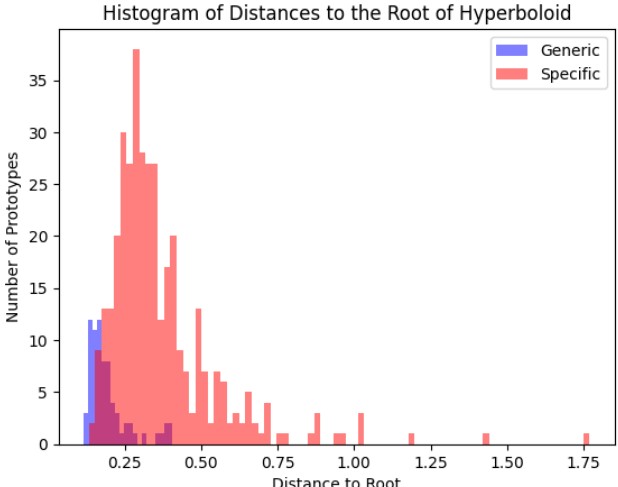

Figure 4. Distribution of distances from prototypes to the origin of the hyperboloid for generic and specific prototypes.

## B. Placement of Prototypes in the Hyperbolic Space

To analyze the distribution of prototypes in hyperbolic space, we measured their distances from the origin of the hyperboloid. Figure 4 shows the distance distributions for generic and specific prototypes in HAPPI, using the XProtoNet [17] backbone, trained end-to-end (E2E) on the PETS dataset [27]. As illustrated, generic prototypes predominantly cluster closer to the origin, reflecting their role in capturing localized, distinctive features. This proximity aligns with our hierarchical organization where generic features are positioned near the origin of the hyperboloid. In contrast, specific prototypes are distributed farther from the root, indicating their role in aggregating broader patterns across larger regions.

## C. Implementation Details

All models were trained using the original configurations presented in their respective papers unless stated otherwise. Below, we detail the specific training setup and modifications made for this study.

### C.1. General Training Setup

For all models, PyTorch [28] was used for training, and Weights and Biases [2] was employed to log and monitor the training process. The experiments were conducted on NVIDIA Tesla V100 GPUs with 32GB of memory. Train-

ing was carried out for 100 epochs with the StepLR learning scheduler, which decays the learning rate by a factor of 0.8 every 5 epochs. Each model used 10 specific prototypes, and HAPPI-based models used 1 generic prototype, with one generic feature being extracted per input for HAPPI. The embedding depth $D$ was set to 512 for all models, matching the depth of the extracted features and prototype vectors.

The optimizer used was Adam for most models, except for ProtoPFormer, where AdamW was used. The learning rate for all models was adjusted according to their original configurations, and all models used a batch size of 64. The training process also involved scaling methods to prevent numerical overflow during the exponential mapping of features to the hyperbolic space, which is further discussed in Section A of the supplementary material.

## C.2. ProtoPNet

For ProtoPNet [3], the loss coefficients were set as follows: $\lambda_{\text{clstr\_g}} = 0.1$, $\lambda_{\text{sep\_g}} = 0.01$, $\lambda_{\text{clstr\_s}} = 0.8$, and $\lambda_{\text{sep\_s}} = 0.08$. The batch size was set to 64. The learning rates were configured as follows: for the backbone ResNet-50 [11] and the last layer fully connected classifier $h(.)$, a learning rate of $1 \times 10^{-4}$ was used, while for the rest of the model, a learning rate of $3 \times 10^{-3}$ was applied. When using HAPPI, the learning rate for the curvature of the hyperbolic space and the scaling factor $\alpha$ was set to $5 \times 10^{-4}$. To train the end-to-end (E2E) version, for both Euclidean and HAPPI versions, we used a uniform learning rate of $1 \times 10^{-4}$ for all parameters.

## C.3. XProtoNet

For XProtoNet, the loss coefficients were the same as ProtoPNet: $\lambda_{\text{clstr\_g}} = 0.1$, $\lambda_{\text{sep\_g}} = 0.01$, $\lambda_{\text{clstr\_s}} = 0.8$, and $\lambda_{\text{sep\_s}} = 0.08$. The batch size was 36 with gradient accumulation steps of 2. The learning rates for the original version were set as follows: for the ResNet-50 backbone and the last layer fully connected classifier $h(.)$, a learning rate of $1 \times 10^{-4}$ was used, and for the rest of the model, the learning rate was $3 \times 10^{-3}$. In the HAPPI version, the learning rate for the curvature of the hyperbolic space and the scaling factor $\alpha$ was set to $5 \times 10^{-4}$. The end-to-end (E2E) version used a uniform learning rate of $1 \times 10^{-4}$ for all parameters.

## C.4. MCPNet

For MCPNet [37], we used their published code repositories and reproduced their method without using the center-crop functionality for the images, as used in their original repository.

## C.5. PipNet

For PipNet [25], we used the same configurations as those presented in their original paper.

## C.6. ST-ProtoPNet

For ST-ProtoPNet [38], the batch size was set to 64, in line with the original paper's configuration.

## C.7. ProtoPFormer

For ProtoPFormer [41], the batch size was set to 64, and we used the AdamW optimizer as specified in the original paper. Instead of the Prototypical Part Concentration (PPC) loss, we implemented our clustering and separation loss functions to better align prototypes in hyperbolic space. The CLS token was used as the generic prototype, while the image tokens were treated as specific prototypes.

## C.8. Black-Box Baselines

For the black-box baseline, the batch size was set to 64, in line with the configurations used for other models.