# OpenReview forum: "HAPPI: Hyperbolic Hierarchical Part Prototypes for Image Recognition"
_thecvf.com/ICCV/2025/Workshop/BEW — BEW 2025 Oral_

### Official Review · Reviewer_nXoy · 2025-07-07
**Well written study on prototypes in hyperbolic space with limitations in evaluation of the hierarchical relationship of the learned prototypes**

**Rating:** 5
**Confidence:** 4

**Review:**

Summary
The authors introduce a novel model architecture that projects Euclidean representations into Lorentz space to obtain Lorentz embeddings. Departing from conventional prototype models that treat inputs independently, their approach leverages Lorentz space projections to capture hierarchical relationships between images, enhancing the structural awareness of the learned representations.


Strengths
- The authors show intention to share code.
- The manuscript is well and clearly written, easy to understand and avoids going into too much mathematical detail that is not needed for implementation anyways
- The manuscript is well placed in literature
- The topic of this study is highly relevant
- Two datasets are used
- The proposed entailment loss introduces a hierarchical structure by ensuring each specific prototype is linked to at least one generic prototype, encouraging more semantically meaningful representations. The improvement in classification results may stem from a positive effect of introducing variation in the prototypes.
- The method is model-agnostic, allowing it to be integrated with various backbone architectures and is not a post-hoc method that requires any finetuning of a model.


Weaknesses
- The main weakness of the study is that the evaluation of the prototypes is limited to a handful examples of quantitative evaluations. This is of course very had as there are no well defined evaluation measures for explanations, but more analysis could have been done regarding the hierarchical organization of the prototypes. The few quantitive examples are hard to interpret and don't really address the hierarchical relationship between the prototypes.
- The proposed loss consists of six terms all with weighing parameter that could be of different scales and impact, which could be a problem in hyper parameter tuning, there is no ablation or sensitivity analysis regarding these loss components
- The claim that specific prototypes capture attributes such as pose, colour, or body type is not clearly supported by the visualisations of their representation in Lorentz space.
- There is no direct comparison of the hierarchical structure between Lorentz and Euclidean spaces, limiting the evaluation of the proposed projection’s effectiveness.
- Missing baseline results using purely Euclidean representations makes it difficult to assess the true benefit of the Lorentz projection.

Minor comments:
- Line 264: the time dimension conventionally comes first in coordinate annotations.
- conventionally the curvature is represented with a "k" not a "c"
- 354: using "t" in V_{t} is confusing as V_{t} represents the time dimension earlier

---

### Official Review · Reviewer_zxFW · 2025-07-07
**A solid extension of prototype models in the hyperbolic domain**

**Rating:** 4
**Confidence:** 3

**Review:**

## Summary

The paper introduces a novel prototypical learning framework that operates in hyperbolic space to better capture hierarchical relationships. After reviewing prototype learning and current state-of-the-art methods—which assign each class a set of learned prototypes for improved accuracy and interpretability—the authors argue that hyperbolic geometry offers a more natural embedding for hierarchies. They extend the standard clustering and separation losses to the hyperbolic domain and incorporate an entailment loss inspired by MERU. Empirical evaluation on the Oxford IIT Pets and CUB-200-2011 datasets fine grained classification demonstrates that the hyperbolic model consistently outperforms its Euclidean counterparts. Finally, ablation studies on the entailment coefficient and the number of prototypes reveal how these factors influence performance.

### Strengths

S1. Extending prototype learning into hyperbolic space represents a well-supported and logically sound strategy, especially for data with inherent hierarchies. The authors offer a lucid theoretical rationale demonstrating why hyperbolic embeddings are better suited to these tasks and empirically show improvements over all Euclidean baselines.

S2. The method demonstrates strong adaptability across diverse architectures, highlighting its versatility and modular design. Additionally, the extensive set of evaluated baselines is commendable.

### Weaknesses

W1. Although the gains over Euclidean baselines are evident, it is surprising that certain network architectures achieve identical performance. Intuitively, even smaller networks should leverage hyperbolic embeddings to more effectively model hierarchical relationships despite their reduced capacity. The statement that more powerful model also have "saturated" representations is not debatable, as some of the cited models (MERU, HycoCLIP, etc) are based on powerful architectures but still benefit from the hyperbolic components.

W2. Even if using identical coefficients for clustering and separation losses is justified, it would be valuable to evaluate the model with distinct coefficient values for each loss term. This exploration could uncover additional performance gains by accounting for the fundamental differences between Euclidean and hyperbolic geometries.

## Questions

In light of weaknesses W1 and W2, it would be helpful to solicit the authors’ perspectives on these observations. Could they elaborate on why certain network architectures fail to benefit from hyperbolic embeddings, and whether they have tested alternative coefficient settings for clustering and separation losses?

Furthermore, the authors might consider adding a future work section to outline avenues for continued development. For example, it would be compelling to apply this approach to larger, naturally hierarchical datasets like TreeOfLife or iNaturalist to assess its scalability and efficacy in more complex settings.

---

### Official Review · Reviewer_RByf · 2025-07-08
**Review of HAPPI: Hyperbolic Hierarchical Part Prototypes for Image Recognition**

**Rating:** 4
**Confidence:** 5

**Review:**

## Summary
This paper aims to address a key shortcoming in part-based prototypical neural networks for image recognition: that all these models treat image features independently and do not consider the hierarchical relationship between them. The paper proposed to train the model in hyperbolic space placing generic features of an image towards the origin and specific features farther away in an entailment relationship. Simpler architectures see significant improvements in performance across two datasets, white advanced architectures see marginal improvements.

## Strengths
- The idea of entailment relationship between generic features and specific features for part-based prototypical networks is novel. It is makes sense intuitively since generic features are common between multiple classes specific features differentiate classes from one another.
- It is encouraging to see good performance improvements across architectures and datasets (CUB, Pets). The improvements are significant on simple architectures like ProtoPNet and marginal on advanced architectures like XProtoNet, ST-ProtoPNet and ProtoPFormer.

## Weaknesses

- The paper focuses on prototypical learning for within-image hierarchy, but does not use the the existing label hierarchy of CUB-200-2011 and Oxford-IIIT Pets. Including label hierarchy during learning can potentially help with separating between classes in a better way while retaining information about part-based prototypes. Of course, in absence of label information, the proposed method already seems to benefit simpler architectures.  Maybe the authors can build upon this in a future work.
- There are no visualizations on interpretability of the method. For example, ProtoPFormer includes visualizations about interpretable decisions of why an image is classified as a _certain bird_. Because the embeddings space learns features in a hierarchical way, it would've been more useful to have a nuanced analysis of how the features affect classification decisions. I'd suggest the authors to include this in camera ready version for completeness.

## Justification
The idea behind the paper is very intuitive. While the performance gain is minimal for complex architectures, the direction of hyperbolic geometry for part-based prototype networks is of value to the broader research in non-euclidean learning. I therefore recommend the paper to be accepted.

---

### Decision · Program_Chairs · 2025-07-09

**Decision:**

Accept (Oral)

**Comment:**

The majority of the reviews agree towards accepting the paper.  The authors should do their best to address the comments of the reviewers in their final version.  The oral presentations would also present a poster.